# Residual Flows for Invertible Generative Modeling

**Ricky T. Q. Chen**[1,3]**, Jens Behrmann**[2]**, David Duvenaud**[1,3]**, Jörn-Henrik Jacobsen**[1,3]
University of Toronto[1], University of Bremen[2], Vector Institute[3]

rtqichen@cs.toronto.edu, jensb@uni-bremen.de
duvenaud@cs.toronto.edu, j.jacobsen@vectorinstitute.ai

## Abstract

Flow-based generative models parameterize probability distributions through an invertible transformation and can be trained by maximum likelihood. Invertible residual networks provide a flexible family of transformations where only Lipschitz conditions rather than strict architectural constraints are needed for enforcing invertibility. However, prior work trained invertible residual networks for density estimation by relying on biased log-density estimates whose bias increased with the network's expressiveness. We give a tractable unbiased estimate of the log density using a "Russian roulette" estimator, and reduce the memory required during training by using an alternative infinite series for the gradient. Furthermore, we improve invertible residual blocks by proposing the use of activation functions that avoid derivative saturation and generalizing the Lipschitz condition to induced mixed norms. The resulting approach, called Residual Flows, achieves state-of-the-art performance on density estimation amongst flow-based models, and outperforms networks that use coupling blocks at joint generative and discriminative modeling.

## 1 Introduction

Maximum likelihood is a core machine learning paradigm that poses learning as a distribution alignment problem. However, it is often unclear what family of distributions should be used to fit high-dimensional continuous data. In this regard, the change of variables theorem offers an appealing way to construct flexible distributions that allow tractable exact sampling and efficient evaluation of its density. This class of models is generally referred to as invertible or flow-based generative models (Deco and Brauer, 1995; Rezende and Mohamed, 2015).

With invertibility as its core design principle, flow-based models (also referred to as normalizing flows) have shown to be capable of generating realistic images (Kingma and Dhariwal, 2018) and can achieve density estimation performance on-par with competing state-of-the-art approaches (Ho et al., 2019). In applications, they have been applied to study adversarial robustness (Jacobsen et al., 2019) and are used to train hybrid models with both generative and classification capabilities (Nalisnick et al., 2019) using a weighted maximum likelihood objective.

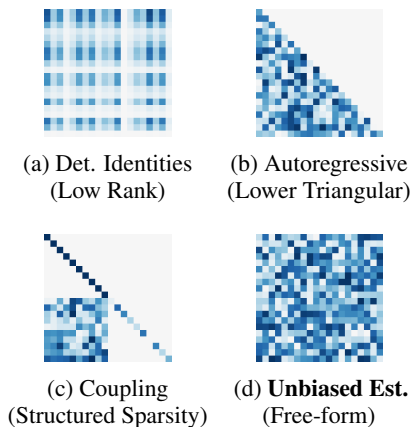

(a) Det. Identities (Low Rank)

(b) Autoregressive (Lower Triangular)

(c) Coupling (Structured Sparsity)

(d) **Unbiased Est.** (Free-form)

Figure 1: **Pathways to designing scalable normalizing flows** and their enforced Jacobian structure. Residual Flows fall under unbiased estimation with free-form Jacobian.

Existing flow-based models (Rezende and Mohamed, 2015; Kingma et al., 2016; Dinh et al., 2014; Chen et al., 2018) make use of restricted transformations with sparse or structured Jacobians (Fig-

ure 1). These allow efficient computation of the log probability under the model but at the cost of architectural engineering. Transformations that scale to high-dimensional data rely on specialized architectures such as coupling blocks (Dinh et al., 2014, 2017) or solving an ordinary differential equation (Grathwohl et al., 2019). Such approaches have a strong inductive bias that can hinder their application in other tasks, such as learning representations that are suitable for both generative and discriminative tasks.

Recent work by Behrmann et al. (2019) showed that residual networks (He et al., 2016) can be made invertible by simply enforcing a Lipschitz constraint, allowing to use a very successful discriminative deep network architecture for unsupervised flow-based modeling. Unfortunately, the density evaluation requires computing an infinite series. The choice of a fixed truncation estimator used by Behrmann et al. (2019) leads to substantial bias that is tightly coupled with the expressiveness of the network, and cannot be said to be performing maximum likelihood as bias is introduced in the objective and gradients.

In this work, we introduce Residual Flows, a flow-based generative model that produces an unbiased estimate of the log density and has memory-efficient backpropagation through the log density computation. This allows us to use expressive architectures and train via maximum likelihood. Furthermore, we propose and experiment with the use of activations functions that avoid derivative saturation and induced mixed norms for Lipschitz-constrained neural networks.

## 2 Background

**Maximum likelihood estimation.** To perform maximum likelihood with stochastic gradient descent, it is sufficient to have an unbiased estimator for the gradient as

$$\nabla_\theta D_{\text{KL}}(p_{\text{data}} \,||\, p_\theta) = \nabla_\theta \mathbb{E}_{x \sim p_{\text{data}}(x)} \left[ \log p_\theta(x) \right] = \mathbb{E}_{x \sim p_{\text{data}}(x)} \left[ \nabla_\theta \log p_\theta(x) \right], \tag{1}$$

where $p_{\text{data}}$ is the unknown data distribution which can be sampled from and $p_\theta$ is the model distribution. An unbiased estimator of the gradient also immediately follows from an unbiased estimator of the log density function, $\log p_\theta(x)$.

**Change of variables theorem.** With an invertible transformation $f$, the change of variables

$$\log p(x) = \log p(f(x)) + \log \left| \det \frac{df(x)}{dx} \right| \tag{2}$$

captures the change in density of the transformed samples. A simple base distribution such as a standard normal is often used for $\log p(f(x))$. Tractable evaluation of (2) allows flow-based models to be trained using the maximum likelihood objective (1). In contrast, variational autoencoders (Kingma and Welling, 2014) can only optimize a stochastic lower bound, and generative adversarial networks (Goodfellow et al., 2014) require an extra discriminator network for training.

**Invertible residual networks (i-ResNets).** Residual networks are composed of simple transformations $y = f(x) = x + g(x)$. Behrmann et al. (2019) noted that this transformation is invertible by the Banach fixed point theorem if $g$ is contractive, i.e. with Lipschitz constant strictly less than unity, which was enforced using spectral normalization (Miyato et al., 2018; Gouk et al., 2018).

Applying i-ResNets to the change-of-variables (2), the identity

$$\log p(x) = \log p(f(x)) + \text{tr} \left( \sum_{k=1}^{\infty} \frac{(-1)^{k+1}}{k} [J_g(x)]^k \right) \tag{3}$$

was shown, where $J_g(x) = \frac{dg(x)}{dx}$. Furthermore, the Skilling-Hutchinson estimator (Skilling, 1989; Hutchinson, 1990) was used to estimate the trace in the power series. Behrmann et al. (2019) used a fixed truncation to approximate the infinite series in (3). However, this naïve approach has a bias that grows with the number of dimensions of $x$ and the Lipschitz constant of $g$, as both affect the convergence rate of this power series. As such, the fixed truncation estimator requires a careful balance between bias and expressiveness, and cannot scale to higher dimensional data. Without decoupling the objective and estimation bias, i-ResNets end up optimizing for the bias without improving the actual maximum likelihood objective (see Figure 2).

# 3 Residual Flows

## 3.1 Unbiased Log Density Estimation for Maximum Likelihood Estimation

Evaluation of the exact log density function $\log p_\theta(\cdot)$ in (3) requires infinite time due to the power series. Instead, we rely on randomization to derive an unbiased estimator that can be computed in finite time (with probability one) based on an existing concept (Kahn, 1955).

To illustrate the idea, let $\Delta_k$ denote the $k$-th term of an infinite series, and suppose we always evaluate the first term then flip a coin $b \sim \text{Bernoulli}(q)$ to determine whether we stop or continue evaluating the remaining terms. By reweighting the remaining terms by $\frac{1}{1-q}$, we obtain an unbiased estimator

$$\Delta_1 + \mathbb{E}\left[\left(\frac{\sum_{k=2}^\infty \Delta_k}{1-q}\right)\mathbb{1}_{b=0} + (0)\mathbb{1}_{b=1}\right] = \Delta_1 + \frac{\sum_{k=2}^\infty \Delta_k}{1-q}(1-q) = \sum_{k=1}^\infty \Delta_k. \tag{4}$$

Interestingly, whereas naïve computation would always use infinite compute, this unbiased estimator has probability $q$ of being evaluated in finite time. We can obtain an estimator that is evaluated in finite time with probability one by applying this process infinitely many times to the remaining terms. Directly sampling the number of evaluated terms, we obtain the appropriately named "Russian roulette" estimator (Kahn, 1955)

$$\sum_{k=1}^\infty \Delta_k = \mathbb{E}_{n\sim p(N)}\left[\sum_{k=1}^n \frac{\Delta_k}{\mathbb{P}(N \geq k)}\right]. \tag{5}$$

We note that the explanation above is only meant to be an intuitive guide and not a formal derivation. The peculiarities of dealing with infinite quantities dictate that we must make assumptions on $\Delta_k$, $p(N)$, or both in order for the equality in (5) to hold. While many existing works have made different assumptions depending on specific applications of (5), we state our result as a theorem where the only condition is that $p(N)$ must have support over all of the indices.

**Theorem 1** (Unbiased log density estimator). *Let $f(x) = x + g(x)$ with $\text{Lip}(g) < 1$ and $N$ be a random variable with support over the positive integers. Then*

$$\log p(x) = \log p(f(x)) + \mathbb{E}_{n,v}\left[\sum_{k=1}^n \frac{(-1)^{k+1}}{k}\frac{v^T[J_g(x)^k]v}{\mathbb{P}(N \geq k)}\right], \tag{6}$$

*where $n \sim p(N)$ and $v \sim \mathcal{N}(0, I)$.*

Here we have used the Skilling-Hutchinson trace estimator (Skilling, 1989; Hutchinson, 1990) to estimate the trace of the matrices $J_g^k$. A detailed proof is given in Appendix B.

Note that since $J_g$ is constrained to have a spectral radius less than unity, the power series converges exponentially. The variance of the Russian roulette estimator is small when the infinite series exhibits fast convergence (Rhee and Glynn, 2015; Beatson and Adams, 2019), and in practice, we did not have to tune $p(N)$ for variance reduction. Instead, in our experiments, we compute two terms exactly and then use the unbiased estimator on the remaining terms with a single sample from $p(N) = \text{Geom}(0.5)$. This results in an expected compute cost of 4 terms, which is less than the 5 to 10 terms that Behrmann et al. (2019) used for their biased estimator.

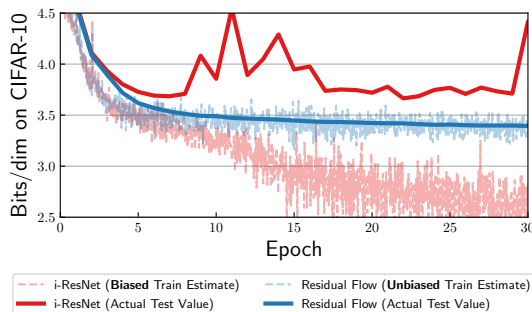

Figure 2: i-ResNets suffer from substantial bias when using expressive networks, whereas Residual Flows principledly perform maximum likelihood with unbiased stochastic gradients.

Theorem 1 forms the core of Residual Flows, as we can now perform maximum likelihood training by backpropagating through (6) to obtain unbiased gradients. This allows us to train more expressive networks where a biased estimator would fail (Figure 2). The price we pay for the unbiased estimator is variable compute and memory, as each sample of the log density uses a random number of terms in the power series.

## 3.2 Memory-Efficient Backpropagation

Memory can be a scarce resource, and running out of memory due to a large sample from the unbiased estimator can halt training unexpectedly. To this end, we propose two methods to reduce the memory consumption during training.

To see how naïve backpropagation can be problematic, the gradient w.r.t. parameters $\theta$ by directly differentiating through the power series (6) can be expressed as

$$\frac{\partial}{\partial \theta} \log \det \left( I + J_g(x, \theta) \right) = \mathbb{E}_{n,v} \left[ \sum_{k=1}^{n} \frac{(-1)^{k+1}}{k} \frac{\partial v^T (J_g(x, \theta)^k) v}{\partial \theta} \right]. \tag{7}$$

Unfortunately, this estimator requires each term to be stored in memory because $\partial/\partial \theta$ needs to be applied to each term. The total memory cost is then $\mathcal{O}(n \cdot m)$ where $n$ is the number of computed terms and $m$ is the number of residual blocks in the entire network. This is extremely memory-hungry during training, and a large random sample of $n$ can occasionally result in running out of memory.

**Neumann gradient series.** Instead, we can specifically express the gradients as a power series derived from a Neumann series (see Appendix C). Applying the Russian roulette and trace estimators, we obtain the following theorem.

**Theorem 2** (Unbiased log-determinant gradient estimator). *Let* $\mathrm{Lip}(g) < 1$ *and $N$ be a random variable with support over positive integers. Then*

$$\frac{\partial}{\partial \theta} \log \det \left( I + J_g(x, \theta) \right) = \mathbb{E}_{n,v} \left[ \left( \sum_{k=0}^{n} \frac{(-1)^k}{\mathbb{P}(N \geq k)} v^T J(x, \theta)^k \right) \frac{\partial (J_g(x, \theta))}{\partial \theta} v \right], \tag{8}$$

*where* $n \sim p(N)$ *and* $v \sim \mathcal{N}(0, I)$.

As the power series in (8) does not need to be differentiated through, using this reduces the memory requirement by a factor of $n$. This is especially useful when using the unbiased estimator as the memory will be constant regardless of the number of terms we draw from $p(N)$.

**Backward-in-forward: early computation of gradients.** We can further reduce memory by partially performing backpropagation during the forward evaluation. By taking advantage of $\log \det(I + J_g(x, \theta))$ being a scalar quantity, the partial derivative from the objective $\mathcal{L}$ is

$$\frac{\partial \mathcal{L}}{\partial \theta} = \underbrace{\frac{\partial \mathcal{L}}{\partial \log \det(I + J_g(x, \theta))}}_{\text{scalar}} \underbrace{\frac{\partial \log \det(I + J_g(x, \theta))}{\partial \theta}}_{\text{vector}}. \tag{9}$$

For every residual block, we compute $\partial \log \det(I+J_g(x,\theta))/\partial \theta$ along with the forward pass, release the memory for the computation graph, then simply multiply by $\partial \mathcal{L}/\partial \log \det(I+J_g(x,\theta))$ later during the main backprop. This reduces memory by another factor of $m$ to $\mathcal{O}(1)$ with negligible overhead.

Note that while these two tricks remove the memory cost from backpropagating through the $\log \det$ terms, computing the path-wise derivatives from $\log p(f(x))$ still requires the same amount of memory as a single evaluation of the residual network. Figure 3 shows that the memory consumption can be enormous for naïve backpropagation, and using large networks would have been intractable.

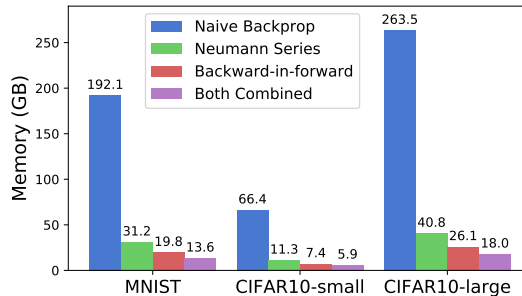

Figure 3: Memory usage (GB) per minibatch of 64 samples when computing $n{=}10$ terms in the corresponding power series. *CIFAR10-small* uses immediate downsampling before any residual blocks.

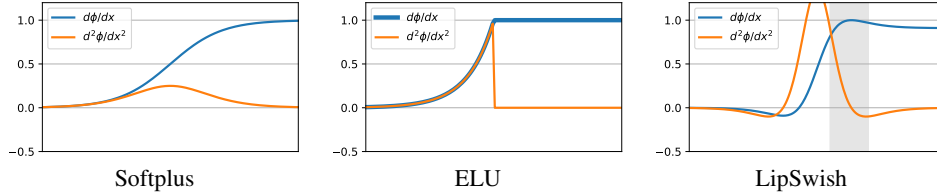

Figure 4: Common smooth Lipschitz activation functions $\phi$ usually have vanishing $\phi''$ when $\phi'$ is maximal. LipSwish has a non-vanishing $\phi''$ in the region where $\phi'$ is close to one.

### 3.3 Avoiding Derivative Saturation with the LipSwish Activation Function

As the log density depends on the first derivatives through the Jacobian $J_g$, the gradients for training depend on second derivatives. Similar to the phenomenon of saturated activation functions, Lipschitz-constrained activation functions can have a derivative saturation problem. For instance, the ELU activation used by Behrmann et al. (2019) achieves the highest Lipschitz constant when $\text{ELU}'(z) = 1$, but this occurs when the second derivative is exactly zero in a very large region, implying there is a trade-off between a large Lipschitz constant and non-vanishing gradients.

We thus desire two properties from our activation functions $\phi(z)$:

1. The first derivatives must be bounded as $|\phi'(z)| \leq 1$ for all $z$
2. The second derivatives should not asymptotically vanish when $|\phi'(z)|$ is close to one.

While many activation functions satisfy condition 1, most do not satisfy condition 2. We argue that the ELU and softplus activations are suboptimal due to derivative saturation. Figure 4 shows that when softplus and ELU saturate at regions of unit Lipschitz, the second derivative goes to zero, which can lead to vanishing gradients during training.

We find that good activation functions satisfying condition 2 are *smooth and non-monotonic* functions, such as Swish (Ramachandran et al., 2017). However, Swish by default does not satisfy condition 1 as $\max_z |\frac{d}{dz}\text{Swish}(z)| \lesssim 1.1$. But scaling via

$$\text{LipSwish}(z) := \text{Swish}(z)/1.1 = z \cdot \sigma(\beta z)/1.1, \tag{10}$$

where $\sigma$ is the sigmoid function, results in $\max_z |\frac{d}{dz}\text{LipSwish}(z)| \leq 1$ for all values of $\beta$. LipSwish is a simple modification to Swish that exhibits a less than unity Lipschitz property. In our experiments, we parameterize $\beta$ to be strictly positive by passing it through softplus. Figure 4 shows that in the region of maximal Lipschitz, LipSwish does not saturate due to its non-monotonicity property.

## 4 Related Work

**Estimation of Infinite Series.** Our derivation of the unbiased estimator follows from the general approach of using a randomized truncation (Kahn, 1955). This paradigm of estimation has been repeatedly rediscovered and applied in many fields, including solving of stochastic differential equations (McLeish, 2011; Rhee and Glynn, 2012, 2015), ray tracing for rendering paths of light (Arvo and Kirk, 1990), and estimating limiting behavior of optimization problems (Tallec and Ollivier, 2017; Beatson and Adams, 2019), among many other applications. Some recent works use Chebyshev polynomials to estimate the spectral functions of symmetric matrices (Han et al., 2018; Adams et al., 2018; Ramesh and LeCun, 2018; Boutsidis et al., 2008). These works estimate quantities that are similar to those presented in this work, but a key difference is that the Jacobian in our power series is not symmetric. We also note works that have rediscovered the random truncation approach (McLeish, 2011; Rhee and Glynn, 2015; Han et al., 2018) made assumptions on $p(N)$ in order for it to be applicable to general infinite series. Fortunately, since the power series in Theorems 1 and 2 converge fast enough, we were able to make use of a different set of assumptions requiring only that $p(N)$ has sufficient support, which was adapted from Bouchard-Côté (2018) (details in Appendix B).

**Memory-efficient Backpropagation.** The issue of computing gradients in a memory-efficient manner was explored by Gomez et al. (2017) and Chang et al. (2018) for residual networks with a

Table 1: Results [bits/dim] on standard benchmark datasets for density estimation. In brackets are models that used "variational dequantization" (Ho et al., 2019), which we don't compare against.

| Model | MNIST | CIFAR-10 | ImageNet 32 | ImageNet 64 | CelebA-HQ 256 |
|---|---|---|---|---|---|
| Real NVP (Dinh et al., 2017) | 1.06 | 3.49 | 4.28 | 3.98 | — |
| Glow (Kingma and Dhariwal, 2018) | 1.05 | 3.35 | 4.09 | 3.81 | 1.03 |
| FFJORD (Grathwohl et al., 2019) | 0.99 | 3.40 | — | — | — |
| Flow++ (Ho et al., 2019) | — | 3.29 (3.09) | — (3.86) | — (3.69) | — |
| i-ResNet (Behrmann et al., 2019) | 1.05 | 3.45 | — | — | — |
| Residual Flow (Ours) | **0.970** | **3.280** | **4.010** | **3.757** | **0.992** |

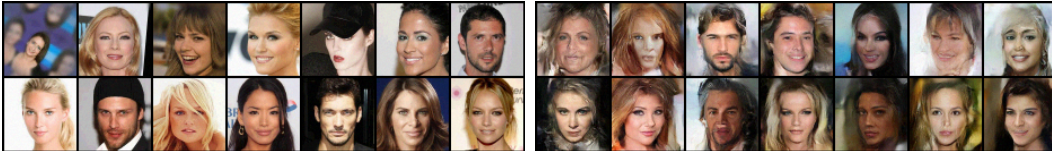

Figure 5: **Qualitative samples.** Real (left) and random samples (right) from a model trained on 5bit 64×64 CelebA. The most visually appealing samples were picked out of 5 random batches.

coupling-based architecture devised by Dinh et al. (2014), and explored by Chen et al. (2018) for a continuous analogue of residual networks. These works focus on the path-wise gradients from the output of the network, whereas we focus on the gradients from the log-determinant term in the change of variables equation specifically for generative modeling. On the other hand, our approach shares some similarities with Recurrent Backpropagation (Almeida, 1987; Pineda, 1987; Liao et al., 2018), since both approaches leverage convergent dynamics to modify the derivatives.

**Invertible Deep Networks.** Flow-based generative models are a density estimation approach which has invertibility as its core design principle (Rezende and Mohamed, 2015; Deco and Brauer, 1995). Most recent work on flows focuses on designing maximally expressive architectures while maintaining invertibility and tractable log determinant computation (Dinh et al., 2014, 2017; Kingma and Dhariwal, 2018). An alternative route has been taken by Continuous Normalizing Flows (Chen et al., 2018) which make use of Jacobian traces instead of Jacobian determinants, provided that the transformation is parameterized by an ordinary differential equation. Invertible architectures are also of interest for discriminative problems, as their information-preservation properties make them suitable candidates for analyzing and regularizing learned representations (Jacobsen et al., 2019).

## 5 Experiments

### 5.1 Density & Generative Modeling

We use a similar architecture as Behrmann et al. (2019), except without the immediate invertible downsampling (Dinh et al., 2017) at the image pixel-level. Removing this substantially increases the amount of memory required (shown in Figure 3) as there are more spatial dimensions at every layer, but increases the overall performance. We also increase the bound on the Lipschitz constants of each weight matrix to 0.98, whereas Behrmann et al. (2019) used 0.90 to reduce the error of the biased estimator. More detailed description of architectures is in Appendix E.

Unlike prior works that use multiple GPUs, large batch sizes, and a few hundred epochs, Residual Flow models are trained with the standard batch size of 64 and converges in roughly 300-350 epochs for MNIST and CIFAR-10. Most network settings can fit on a single GPU (see Figure 3), though we use 4 GPUs in our experiments to speed up training. On CelebA-HQ, Glow had to use a batchsize of 1 per GPU with a budget of 40 GPUs whereas we trained our model using a batchsize of 3 per GPU and a budget of 4 GPUs, owing to the smaller model and memory-efficient backpropagation.

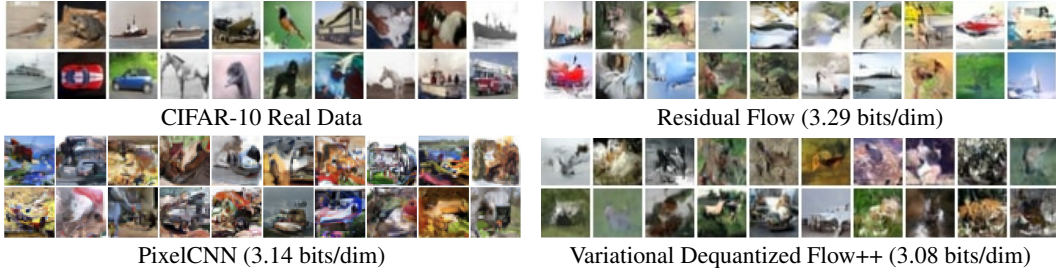

CIFAR-10 Real Data               Residual Flow (3.29 bits/dim)

PixelCNN (3.14 bits/dim)          Variational Dequantized Flow++ (3.08 bits/dim)

Figure 6: Random samples from Residual Flow are more globally coherent. PixelCNN (Oord et al., 2016) and Flow++ samples reprinted from Ho et al. (2019).

Table 1 reports the bits per dimension ($\log_2 p(x)/d$ where $x \in \mathbb{R}^d$) on standard benchmark datasets MNIST, CIFAR-10, downsampled ImageNet, and CelebA-HQ. We achieve competitive performance to state-of-the-art flow-based models on all datasets. For evaluation, we computed 20 terms of the power series (3) and use the unbiased estimator (6) to estimate the remaining terms. This reduces the standard deviation of the unbiased estimate of the test bits per dimension to a negligible level.

Furthermore, it is possible to generalize the Lipschitz condition of Residual Flows to arbitrary p-norms and even mixed matrix norms. By learning the norm orders jointly with the model, we achieved a small gain of 0.003 bits/dim on CIFAR-10 compared to spectral normalization. In addition, we show that others norms like $p = \infty$ yielded constraints more suited for lower dimensional data. See Appendix D for a discussion on how to generalize the Lipschitz condition and an exploration of different norm-constraints for 2D problems and image data.

## 5.2 Sample Quality

We are also competitive with state-of-the-art flow-based models in regards to sample quality. Figure 5 shows random samples from the model trained on CelebA. Furthermore, samples from Residual Flow trained on CIFAR-10 are more globally coherent (Figure 6) than PixelCNN and variational dequantized Flow++, even though our likelihood is worse.

For quantitative comparison, we report FID scores (Heusel et al., 2017) in Table 2. We see that Residual Flows significantly improves on i-ResNets and PixelCNN, and achieves slightly better sample quality than an official Glow model that has double the number of layers. It is well-known that visual fidelity and log-likelihood are not necessarily indicative of each other (Theis et al., 2015), but we believe residual blocks may have a better inductive bias than coupling blocks or autoregressive architectures as generative models. More samples are in Appendix A.

Table 2: Lower FID implies better sample quality. *Results taken from Ostrovski et al. (2018).

| Model | CIFAR10 FID |
|---|---|
| PixelCNN* | 65.93 |
| PixelIQN* | 49.46 |
| i-ResNet | 65.01 |
| Glow | 46.90 |
| Residual Flow | **46.37** |
| DCGAN* | 37.11 |
| WGAN-GP* | 36.40 |

To generate visually appealing images, Kingma and Dhariwal (2018) used temperature annealing (ie. sampling from $[p(x)]^{T^2}$ with $T < 1$) to sample closer to the mode of the distribution, which helped remove artifacts from the samples and resulted in smoother looking images. However, this is done by reducing the entropy of $p(z)$ during sampling, which is only equivalent to temperature annealing if the change in log-density does not depend on the sample itself. Intuitively, this assumption implies that the mode of $p(x)$ and $p(z)$ are the same. As this assumption breaks for general flow-based models, including Residual Flows, we cannot use the same trick to sample efficiently from a temperature annealed model. Figure 7 shows the results of reduced entropy sampling on CelebA-HQ 256, but the samples do not converge to the mode of the distribution.

## 5.3 Ablation Experiments

We report ablation experiments for the unbiased estimator and the LipSwish activation function in Table 3. Even in settings where the Lipschitz constant and bias are relatively low, we observe a significant improvement from using the unbiased estimator. Training the larger i-ResNet model

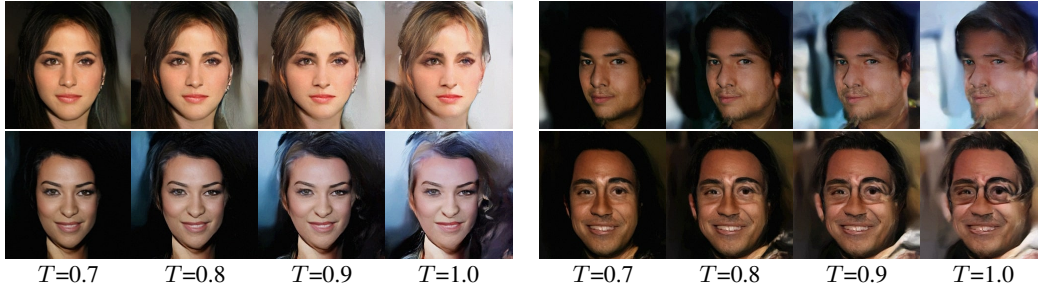

| $T$=0.7 | $T$=0.8 | $T$=0.9 | $T$=1.0 | | $T$=0.7 | $T$=0.8 | $T$=0.9 | $T$=1.0 |

Figure 7: Reduced entropy sampling does not equate with proper temperature annealing for general flow-based models. Naïvely reducing entropy results in samples that exhibit black hair and background, indicating that samples are not converging to the mode of the distribution.

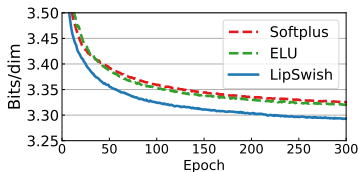

| Training Setting | MNIST | CIFAR-10$^\dagger$ | CIFAR-10 |
|---|---|---|---|
| i-ResNet + ELU | 1.05 | 3.45 | 3.66~4.78 |
| Residual Flow + ELU | 1.00 | 3.40 | 3.32 |
| Residual Flow + LipSwish | **0.97** | **3.39** | **3.28** |

Figure 8: Effect of activation functions on CIFAR-10.

Table 3: Ablation results. $^\dagger$Uses immediate downsampling before any residual blocks.

on CIFAR-10 results in the biased estimator completely ignoring the actual likelihood objective altogether. In this setting, the biased estimate was lower than 0.8 bits/dim by 50 epochs, but the actual bits/dim wildly oscillates above 3.66 bits/dim and seems to never converge. Using LipSwish not only converges much faster but also results in better performance compared to softplus or ELU, especially in the high Lipschitz settings (Figure 8 and Table 3).

## 5.4 Hybrid Modeling

Next, we experiment on joint training of continuous and discrete data. Of particular interest is the ability to learn both a generative model and a classifier, referred to as a hybrid model which is useful for downstream applications such as semi-supervised learning and out-of-distribution detection (Nalisnick et al., 2019). Let $x$ be the data and $y$ be a categorical random variable. The maximum likelihood objective can be separated into $\log p(x,y) = \log p(x) + \log p(y|x)$, where $\log p(x)$ is modeled using a flow-based generative model and $\log p(y|x)$ is a classifier network that shares learned features from the generative model. However, it is often the case that accuracy is the metric of interest and log-likelihood is only used as a surrogate training objective. In this case, (Nalisnick et al., 2019) suggests a weighted maximum likelihood objective,

$$\mathbb{E}_{(x,y)\sim p_{\text{data}}}[\lambda \log p(x) + \log p(y|x)], \tag{11}$$

where $\lambda$ is a scaling constant. As $y$ is much lower dimensional than $x$, setting $\lambda < 1$ emphasizes classification, and setting $\lambda = 0$ results in a classification-only model which can be compared against.

Table 4: Comparison of residual vs. coupling blocks for the hybrid modeling task.

| | MNIST | | | | | | SVHN | | | | | |
|---|---|---|---|---|---|---|---|---|---|---|---|---|
| | $\lambda = 0$ | $\lambda = 1/D$ | | $\lambda = 1$ | | | $\lambda = 0$ | $\lambda = 1/D$ | | $\lambda = 1$ | | |
| Block Type | Acc↑ | BPD↓ | Acc↑ | BPD↓ | Acc↑ | | Acc↑ | BPD↓ | Acc↑ | BPD↓ | Acc↑ | |
| Nalisnick et al. (2019) | 99.33% | 1.26 | 97.78% | – | – | | 95.74% | 2.40 | 94.77% | – | – | |
| Coupling | 99.50% | 1.18 | 98.45% | 1.04 | 95.42% | | 96.27% | 2.73 | 95.15% | 2.21 | 46.22% | |
| + 1 × 1 Conv | **99.56%** | 1.15 | 98.93% | 1.03 | 94.22% | | **96.72%** | 2.61 | 95.49% | 2.17 | 46.58% | |
| Residual | 99.53% | **1.01** | 99.46% | **0.99** | **98.69%** | | **96.72%** | **2.29** | **95.79%** | **2.06** | **58.52%** | |

Since Nalisnick et al. (2019) performs approximate Bayesian inference and uses a different architecture than us, we perform our own ablation experiments to compare residual blocks to coupling blocks (Dinh et al., 2014) as well as $1\times 1$ convolutions (Kingma and Dhariwal, 2018). We use the same architecture as the density estimation experiments and append a classification branch that takes features at the final output of

Table 5: Hybrid modeling results on CIFAR-10.

| Block Type | $\lambda = 0$ Acc↑ | $\lambda = 1/D$ BPD↓ | $\lambda = 1/D$ Acc↑ | $\lambda = 1$ BPD↓ | $\lambda = 1$ Acc↑ |
|---|---|---|---|---|---|
| Coupling | 89.77% | 4.30 | 87.58% | 3.54 | 67.62% |
| + $1 \times 1$ Conv | 90.82% | 4.09 | 87.96% | 3.47 | 67.38% |
| Residual | **91.78%** | **3.62** | **90.47%** | **3.39** | **70.32%** |

multiple scales (see details in Appendix E). This allows us to also use features from intermediate blocks whereas Nalisnick et al. (2019) only used the final output of the entire network for classification. Our implementation of coupling blocks uses the same architecture for $g(x)$ except we use ReLU activations and no longer constrain the Lipschitz constant.

Tables 4 & 5 show our experiment results. Our architecture outperforms Nalisnick et al. (2019) on both pure classification and hybrid modeling. Furthermore, on MNIST we are able to jointly obtain a decent classifier and a strong density model over all settings. In general, we find that residual blocks perform much better than coupling blocks at learning representations for both generative and discriminative tasks. Coupling blocks have very high bits per dimension when $\lambda = 1/D$ while performing worse at classification when $\lambda = 1$, suggesting that they have restricted flexibility and can only perform one task well at a time.

## 6    Conclusion

We have shown that invertible residual networks can be turned into powerful generative models. The proposed unbiased flow-based generative model, coined Residual Flow, achieves competitive or better performance compared to alternative flow-based models in density estimation, sample quality, and hybrid modeling. More generally, we gave a recipe for introducing stochasticity in order to construct tractable flow-based models with a different set of constraints on layer architectures than competing approaches, which rely on exact log-determinant computations. This opens up a new design space of expressive but Lipschitz-constrained architectures that has yet to be explored.

## Acknowledgments

Jens Behrmann gratefully acknowledges the financial support from the German Science Foundation for RTG 2224 "$\pi^3$: Parameter Identification - Analysis, Algorithms, Applications"

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
