[Supplementary Material]

# A  Random Samples

Figure 6: Random samples from CIFAR-10 models. PixelCNN (Oord et al., 2016) and Flow++ samples reprinted from Ho et al. (2019), with permission.

Figure 7: Random samples from MNIST.

Figure 8: Random samples from ImageNet 32×32.

Real Data                                                        Residual Flow

Figure 9: Random samples from ImageNet 64×64.

Real Data                                                        Residual Flow

Figure 10: Random samples from 5bit CelebA 64×64.

## B  Proofs

We start by formulating a Lemma, which gives the condition when the randomized truncated series is an unbiased estimator in a fairly general setting. Afterwards, we study our specific estimator and prove that the assumption of the Lemma is satisfied.

Note, that similar conditions have been stated in previous works, e.g. in McLeish (2011) and Rhee and Glynn (2012). However, we use the condition from (Bouchard-Côté, 2018), which only requires $p(N)$ to have sufficient support.

To make the derivations self-contained, we reformulate the conditions from (Bouchard-Côté, 2018) in the following way:

**Lemma 3** (Unbiased randomized truncated series)**.** *Let $Y_k$ be a real random variable with* $\lim_{k \to \infty} \mathbb{E}[Y_k] = a$ *for some* $a \in \mathbb{R}$*. Further, let $\Delta_0 = Y_0$ and $\Delta_k = Y_k - Y_{k-1}$ for $k \geq 1$.*

*Assume*

$$\mathbb{E}\left[\sum_{k=0}^{\infty}|\Delta_k|\right] < \infty$$

*and let $N$ be a random variable with support over the positive integers and $n \sim p(N)$. Then for*

$$Z = \sum_{k=0}^{n}\frac{\Delta_k}{\mathbb{P}(N \geq k)},$$

*it holds*

$$a = \lim_{k \to \infty}\mathbb{E}[Y_k] = \mathbb{E}_{n \sim p(N)}[Z] = a.$$

*Proof.* First, denote

$$Z_M = \sum_{k=0}^{M}\frac{\mathbb{1}[N \geq k]\Delta_k}{\mathbb{P}(N \geq k)} \quad \text{and} \quad B_M = \sum_{k=0}^{M}\frac{\mathbb{1}[N \geq k]|\Delta_k|}{\mathbb{P}(N \geq k)},$$

where $|Z_M| \leq B_M$ by the triangle inequality. Since $B_M$ is non-decreasing, the monotone convergence theorem allows swapping the expectation and limit as $\mathbb{E}[B] = \mathbb{E}[\lim_{M \to \infty} B_M] = \lim_{M \to \infty}\mathbb{E}[B_M]$. Furthermore, it is

$$\mathbb{E}[B] = \lim_{M \to \infty}\mathbb{E}[B_M] = \lim_{M \to \infty}\sum_{k=0}^{M}\mathbb{E}\left[\frac{\mathbb{1}[N \geq k]|\Delta_k|}{\mathbb{P}(N \geq k)}\right]$$

$$= \lim_{M \to \infty}\sum_{k=0}^{M}\frac{\mathbb{P}(N \geq k)\mathbb{E}|\Delta_k|}{\mathbb{P}(N \geq k)} = \mathbb{E}\left[\lim_{M \to \infty}\sum_{k=0}^{M}|\Delta_k|\right] < \infty,$$

where the assumption is used in the last step. Using the above, the dominated convergence theorem can be used to swap the limit and expectation for $Z_M$. Using similar derivations as above, it is

$$\mathbb{E}[Z] = \lim_{M \to \infty}\mathbb{E}[Z_M] = \lim_{M \to \infty}\mathbb{E}\left[\sum_{k=0}^{M}\Delta_k\right] = \lim_{M \to \infty}\mathbb{E}[Y_k] = a,$$

where we used the definition of $Y_M$ and $\lim_{k \to \infty}\mathbb{E}[Y_k] = a$. $\qquad\square$

*Proof.* **(Theorem 1)**
To simplify notation, we denote $J := J_g(x)$. Furthermore, let

$$Y_N = \mathbb{E}_v\left[\sum_{k=1}^{N}\frac{(-1)^{k+1}}{k}v^T J^k v\right]$$

denote the real random variable and let $\Delta_0 = Y_0$ and $\Delta_k = Y_k - Y_{k-1}$ for $k \geq 1$, as in Lemma 3. To prove the claim of the theorem, we can use Lemma 3 and we only need to prove that the assumption $\mathbb{E}_v[\sum_{k=1}^{\infty}|\Delta_k|] < \infty$ holds for this specific case.

In order to exchange summation and expectation via Fubini's theorem, we need to proof that $\sum_{k=1}^{\infty}|\Delta_k| < \infty$ for all $v \sim \mathcal{N}(0, I)$. Using the assumption $\mathrm{Lip}(g) < 1$, it is

$$\sum_{k=1}^{\infty}|\Delta_k| = \sum_{k=1}^{\infty}\left|\frac{(-1)^{k+1}}{k}v^T J^k v\right| = \sum_{k=1}^{\infty}\frac{\|v^T J^k v\|_2}{k} \leq \sum_{k=1}^{\infty}\frac{\|v^T\|_2\|J^k\|_2\|v\|_2}{k}$$

$$\leq 2\|v\|_2\sum_{k=1}^{\infty}\frac{\|J\|_2^k}{k} \leq 2\|v\|_2\sum_{k=1}^{\infty}\frac{\mathrm{Lip}(g)_2^k}{k} = 2\|v\|_2\log\left(1 - \mathrm{Lip}(g)\right) < \infty,$$

for an arbitrary $v$. Hence,

$$\mathbb{E}_v\left[\sum_{k=1}^{\infty}|\Delta_k|\right] = \sum_{k=1}^{\infty}\mathbb{E}_v[|\Delta_k|]. \tag{13}$$

403 Since $\mathrm{tr}(A) = \mathbb{E}_v[v^T A v]$ for $v \sim \mathcal{N}(0, I)$ via the Skilling-Hutchinson trace estimator (Hutchinson,
404 1990; Skilling, 1989), it is

$$\mathbb{E}_v[|\Delta_k|] = \left| \frac{\mathrm{tr}(J^k)}{k} \right|.$$

405 To show that (13) is bounded, we derive the bound

$$\frac{1}{k} |\mathrm{tr}(J^k)| \leq \frac{1}{k} \left| \sum_{i=d}^{d} \lambda_i(J^k) \right| \leq \frac{1}{k} \sum_{i=d}^{d} |\lambda_i(J^k)| \leq \frac{d}{k} \rho(J^k) \leq \frac{d}{k} \|J^k\|_2 \leq \frac{d}{k} \mathrm{Lip}(g)^k,$$

406 where $\lambda(J^k)$ denote the eigenvalues and $\rho(J^k)$ the spectral radius. Inserting this bound into (13) and
407 using $\mathrm{Lip}(g) < 1$ yields

$$\mathbb{E}_v[|\Delta_k|] \leq d \sum_{k=1}^{\infty} \frac{\mathrm{Lip}(g)^k}{k} = -d \log \left( 1 - \mathrm{Lip}(g) \right) < \infty.$$

408 Hence, the assumption of Lemma 3 is verified. □

409 *Proof.* **(Theorem 2)**
410 The result can be proven in an analogous fashion to the proof of Theorem 1, which is why we only
411 present a short version without all steps.

412 By obtaining the bound

$$\sum_{k=0}^{\infty} \left| (-1)^k v^T \left( J(x, \theta)^k \frac{\partial(J_g(x, \theta))}{\partial \theta} \right) v \right| \leq 2\|v\|_2 \left\| \frac{\partial(J_g(x, \theta))}{\partial \theta} \right\| \sum_{k=0}^{\infty} \mathrm{Lip}(g)^k$$
$$= 2\|v\|_2 \left\| \frac{\partial(J_g(x, \theta))}{\partial \theta} \right\| \frac{1}{1 - \mathrm{Lip}(g)} < \infty,$$

413 Fubini's theorem can be applied to swap the expection and summation. Furthermore, by using the
414 trace estimation and similar bounds as in the proof of Theorem 1, the assumption $\mathbb{E}\left[ \sum_{k=0}^{\infty} |\Delta_k| \right] < \infty$
415 from Lemma 3 can be proven.

416 □

## C   Memory-Efficient Gradient Estimation of Log-Determinant

418 Derivation of gradient estimator via differentiating power series:

$$\frac{\partial}{\partial \theta_i} \log \det \left( I + J_g(x, \theta) \right) = \frac{\partial}{\partial \theta_i} \left( \sum_{k=1}^{\infty} (-1)^{k+1} \frac{\mathrm{tr}(J_g(x, \theta)^k)}{k} \right)$$
$$= \mathrm{tr} \left( \sum_{k=1}^{\infty} \frac{(-1)^{k+1}}{k} \frac{\partial(J_g(x, \theta)^k)}{\partial \theta_i} \right)$$

Figure 11: **Learned norm orders on CIFAR-10.** Each residual block is visualized as a single line. The input and two hidden states for each block use different normed spaces. We observe multiple trends: (i) the norms for the first hidden states are consistently higher than the input, and lower for the second. (ii) The orders for the hidden states drift farther away from 2 as depth increases. (iii) The ending order of one block and the starting order of the next are generally consistent and close to 2.

Derivation of memory-efficient gradient estimator:

$$\frac{\partial}{\partial \theta_i} \log \det \left( I + J_g(x, \theta) \right)$$

$$= \frac{1}{\det(I + J_g(x, \theta))} \left[ \frac{\partial}{\partial \theta_i} \det \left( I + J_g(x, \theta) \right) \right] \tag{14}$$

$$= \frac{1}{\det(I + J_g(x, \theta))} \left[ \det(I + J_g(x, \theta)) \operatorname{tr} \left( (I + J(x, \theta))^{-1} \frac{\partial (I + J_g(x, \theta))}{\partial \theta_i} \right) \right] \tag{15}$$

$$= \operatorname{tr} \left( (I + J(x, \theta))^{-1} \frac{\partial (I + J_g(x, \theta))}{\partial \theta_i} \right)$$

$$= \operatorname{tr} \left( (I + J(x, \theta))^{-1} \frac{\partial (J_g(x, \theta))}{\partial \theta_i} \right)$$

$$= \operatorname{tr} \left( \left[ \sum_{k=0}^{\infty} (-1)^k J(x, \theta)^k \right] \frac{\partial (J_g(x, \theta))}{\partial \theta_i} \right) \tag{16}$$

Note, that (14) follows from the chain rule of differentiation, for the derivative of the determinant in (15), see (Petersen and Pedersen, 2012) (eq. 46). Furthermore, (16) follows from the properties of a Neumann-Series which converges due to $\|J_g(x, \theta)\| < 1$.

Hence, if we are able to compute the trace exactly, both approaches will return the same values for a given truncation $n$. However, when estimating the trace via the Hutchinson trace estimator the estimation is not equal in general:

$$v^T \left( \sum_{k=1}^{\infty} \frac{(-1)^{k+1}}{k} \frac{\partial (J_g(x, \theta)^k)}{\partial \theta_i} \right) v \neq v^T \left( \left[ \sum_{k=0}^{\infty} (-1)^k J_g^k(x, \theta) \right] \frac{\partial (J_g(x, \theta))}{\partial \theta_i} \right) v.$$

Another difference between both approaches is their memory consumption of the corresponding computational graph. The summation $\sum_{k=0}^{\infty} (-1)^k J_g^k(x, \theta)$ is not being tracked for the gradient, which allows to compute the gradient with constant memory (constant with respect to the truncation $n$).

# D Generalized Spectral Norm

|  Data | $p = 2$ (5.13 bits) | $p = \infty$ (5.09 bits) |

Figure 12: Learned densities on Checkerboard 2D.

**Using different induced $p$-norms on Checkerboard 2D.**    We experimented with the checkerboard 2D dataset, which is a rather difficult two-dimensional data to fit a flow-based model on due to the discontinuous nature of the true distribution. We used brute-force computation of the log-determinant for change of variables (which, in the 2D case, is faster than the unbiased estimator). In the 2D case, we found that $\infty$-norm always outperforms or at least matches the $p = 2$ norm (ie. spectral norm). Figure 12 shows the learned densities with 200 residual blocks. The color represents the magnitude of $p_\theta(x)$, with brighter values indicating larger values. The $\infty$-norm model produces density estimates that are more evenly spread out across the space, whereas the spectral norm model focused its density to model between-density regions.

Figure 13: Improvement from using generalized spectral norm on CIFAR-10.

**Learning norm orders on CIFAR-10.**    We used $1 + \tanh(s)/2$ where $s$ is a learned weight. This bounds the norm orders to $(1.5, 2.5)$. We tried two different setups. One where all norm orders are free to change (conditioned on them satisfying the constraints (11)), and another setting where each states within each residual block share the same order. Figure 13 shows the improvement in bits from using learned norms. The gain in performance is marginal, and the final models only outperformed spectral norm by around $0.003$ bits/dim. Interestingly, we found that the learned norms stayed around $p = 2$, shown in Figure 11, especially for the input and output spaces of $g$, ie. between blocks. This may suggest that spectral norm, or a norm with $p = 2$ is already optimal in this setting.

# E    Experiment Setup

We use the standard setup of passing the data through a "unsquashing" layer (we used the logit transform (Dinh et al., 2017)), followed by alternating multiple blocks and squeeze layers (Dinh et al., 2017). We use activation normalization (Kingma and Dhariwal, 2018) before and after every residual block. Each residual connection consists of

$$\text{LipSwish} \to 3{\times}3 \text{ Conv} \to \text{LipSwish} \to 1{\times}1 \text{ Conv} \to \text{LipSwish} \to 3{\times}3 \text{ Conv}$$

with hidden dimensions of $512$. Below are the architectures for each dataset.

**MNIST.**    With $\alpha =$1e-5.

$$\text{Image} \to \text{LogitTransform}(\alpha) \to 16{\times}\text{ResBlock} \to \big[\, \text{Squeeze} \to 16{\times}\text{ResBlock} \,\big]{\times}2$$

**CIFAR-10.**    With $\alpha = 0.05$.

$$\text{Image} \to \text{LogitTransform}(\alpha) \to 16{\times}\text{ResBlock} \to \big[\, \text{Squeeze} \to 16{\times}\text{ResBlock} \,\big]{\times}2$$

**ImageNet 32$\times$32.**    With $\alpha = 0.05$.

$$\text{Image} \to \text{LogitTransform}(\alpha) \to 32{\times}\text{ResBlock} \to \big[\, \text{Squeeze} \to 32{\times}\text{ResBlock} \,\big]{\times}2$$

**ImageNet 64$\times$64.**    With $\alpha = 0.05$.

$$\text{Image} \to \text{Squeeze} \to \text{LogitTransform}(\alpha) \to 32{\times}\text{ResBlock} \to \big[\, \text{Squeeze} \to 32{\times}\text{ResBlock} \,\big]{\times}2$$

463 **CelebA 64×64.** With $\alpha = 0.05$.

464      Image → Squeeze → LogitTransform($\alpha$) → 16×ResBlock → $\left[$ Squeeze → 16×ResBlock $\right]$×3

465 For hybrid modeling on CIFAR-10, we replaced the logit transform with normalization by the
466 standard preprocessing of subtracting the mean and dividing by the standard deviation across the
467 training data. The MNIST and SVHN architectures for hybrid modeling were the same as those for
468 density modeling.

469 For augmenting our flow-based model with a classifier in the hybrid modeling experiments, we added
470 an additional branch after every squeeze layer and at the end of the network. Each branch consisted
471 of

472           3×3 Conv → ActNorm → ReLU → AdaptiveAveragePooling($(1, 1)$)

473 where the adaptive average pooling averages across all spatial dimensions and resulted in a vector of
474 dimension 256. The outputs at every scale were concatenated together and fed into a linear softmax
475 classifier.

476 **Adaptive number of power iterations.** To account for variable weight updates during training,
477 we used an adaptive version of spectral normalization for convolutional layers (Gouk et al., 2018)
478 where we performed as many iterations as needed until the relative change in the estimated spectral
479 norm was sufficiently small. As this also reduced the number of iterations when no weight changes
480 occur, it resulted in speed comparable to always performing 5 iterations of power method.

481 **Optimization.** For stochastic gradient descent, we used Adam (Kingma and Ba, 2014) with a
482 learning rate of 0.001 with otherwise default hyperparameters. We used Polyak averaging (Polyak
483 and Juditsky, 1992) for evaluation with a momentum of 0.999.

484 **Preprocessing.** For density estimation experiments, we used random horizontal flipping for CIFAR-
485 10 and CelebA.

486 For hybrid modeling and classification experiments, we used random cropping after reflection padding
487 with 4 pixels for SVHN and CIFAR-10. CIFAR-10 also included random horizontal flipping.