[Reviews · NeurIPS 2019]

Reviewer 1



* Quality Overall the paper is of high quality. My main remarks are: 1. A priori, the hybrid modeling task seems out of place for this paper. I have rarely seen generative models being evaluated on such a task, where the classifier is trained simultaneously as the generative model. Generally there’s some semi-supervised setup where the representations are learned, frozen and used to test their ability to model p(y | x). I can see that one could argue that the inductive bias of residual networks might be more suitable for discriminative tasks as well. It would be good to state the intention of the experiment in the context of the paper more clearly (currently there’s only L251-L252 which is quite vague). 2. The samples in Figure 3 don’t look as competitive to e.g. the samples from Glow (in terms of sample quality). Since I find no mention of temperature scaling, my assumption would be that this was not used. As also mentioned in the Glow paper, temperature scaling is important to get more visually appealing samples (Glow e.g. uses T=0.7). Since this is a generative modeling paper on images, I would suggest to explore temperature scaling and choose a temperature that works best (and report it as such). It would also be important to then show the effect of temperature scaling on the samples in the appendix (Glow has this in the main paper). If this results in much more competitive results, consider moving some of the generalization to the induced mixed norms or the hybrid modeling task to the appendix and putting more emphasis (potentially with a figure on the first page) on the samples. Currently the paper is relatively technical and it would be great to balance that out more (in the context of an improved generative model of images). 3. For CelebA (is it CelebA-HQ?) there are no quantitative results anywhere (just samples).They should ideally also be reported in Table 2. * Originality The related work section seems thorough and the positioning of the paper is very clear. It is clear how the paper overcomes certain limitations in previously introduced methods of interest to the field. * Clarity The paper is well-written, well-structured and reads fluently. * Significance There is a good argument for why invertible residual networks are interesting to people in the field of generative modeling / flow-based modeling, as also explained in the introduction of the paper: it uses a common supervised network architecture (ResNet) and transforms it to one that is tractable for flow-based modeling without (big) changes to the architecture. This seems to be a promising approach compared to previous methods which rely on specialized architectures (coupling blocks, etc.). This paper makes this approach more tractable and also shows quantitative improvements, even compared to the state of the art on some of the tasks. I consider this paper of significance to the field of generative modeling. The samples are less impressive but I also address as to why this could be above. --------------------------- I thank the authors for their response. I will leave my already high score as is; the addressed comments will improve the paper but not significantly.

Reviewer 2



I appreciate the authors' response about generalization beyond normalizing flows. I would encourage the authors to add these generalizations in the conclusion or discussion section to help readers see how these results could be more broadly useful. ---- Original review ---- The novelty primarily comes from deriving an unbiased log density and log determinant estimator rather than using a fixed truncation of an infinite sum, which can be significantly biased. The tradeoff (which the paper mentions clearly) is variable memory and computation during training. Overall, I think this is a focused but solid contribution and is explained clearly. The induced mixed norm section could be shortened to a paragraph and the details left in the appendix. I appreciate including the somewhat negative result but I think a paragraph of explanation and details in the appendix would do fine. Overall, I didn't find any major weaknesses in the paper. One small weakness may be that this primarily builds off of a previous method and improves one particular part of the estimation process. This is definitely useful but the novelty doesn't open up entirely novel models or algorithms and it is not clear that this can be generalized to other related situations (i.e. can these ideas be used for other models/methods than invertible residual flows). Could you mention what distribution you used for p(N)? I might have just missed it but wanted to double check. Also, since any distribution with support on the positive indices allows for unbiasedness, why not choose one that almost always selects a very small number (i.e. Poisson distribution with lambda close to 0)? Could you give some intuitions on what are reasonable values for p(N)? Figure 1 should probably be moved closer to the first mention of it. I didn't understand reading till near the end of the introduction.

Reviewer 3



While the paper is structured as a grab bag of improvements to the i-ResNet model, the methods are original and well explained via text and ablation studies. The work will be helpful for increasing the maturity of these promising residual models. Overall the quality and clarity of the paper is good, although I think there are a couple points in the paper that could use elaboration/clarification. Regarding the inequality in the appendix following equation 16, I presume the reason why these are not equal is because the Jacobian matrix and it's derivative might not commute? Maybe it would be worth mentioning this. If these are indeed not equal, have you observed a difference in the variance of the estimator going from one form to the other (irrespective of the difference in memory costs)? For the backward in forward, it's not clear to me where the memory savings are coming from. It is mentioned that precomputing the log determinant derivative reduces memory (used in logdet computation) from O(m) to O(1) where m is the number of residual blocks. It seems like if these are stored during the forwards pass, then the memory usage for doing so will still be O(m). I think section 3.3 could use a little elaboration on why the LipSwish activation function better satisfies (2) than the softplus function. For the minibatching, presumably the same N is chosen for all elements in the minibatch but I didn't see this mentioned in the paper. The appendix mentions that the number of iterations used in power iteration is adapted so that relative change in estimated norm is small, but doesn't state the exact value. This detail would be important to have.

[Author Response · NeurIPS 2019]

**Aggregated Responses:**  We thank all reviewers for their insightful comments and useful suggestions. Below, we have taken your suggestions which have strengthened the paper.

**FID score for sample quality.**  Reviewers 1 & 3 both suggested reporting FID scores. Our model achieves 46.16 on CIFAR10. For comparison, [2] reports 36.4, 37.11 and 65.93 for WGAN-GP, DCGAN and PixelCNN (lower is better), respectively. We also tested the official Glow model which got a FID score of 46.90, slightly worse than ours.

**CelebAHQ 256x256.**  We have additional experiments on 256x256 images where we achieve 1.00 bits/dim (Glow reports 1.03 bits/dim while using a model double the size of ours). Our samples are similar to Glow's non-temperature annealed samples. However, we note that while Glow used 40 GPUs (with 1 example per GPU and gradient checkpointing), we could train with 3 examples per GPU and a budget of only 4 GPUs.

**Reproducibility.**  We plan on releasing the weights for trained models to allow easier adoption in future works. For instance, this will allow easy computation of various metrics for evaluating sample quality, if needed.

## Reviewer #1:

**Hybrid modeling experiments.**  One of the main motivations behind Residual Flows was to create a family of models which is strong in both discriminative and generative tasks. Hybrid density modeling seemed like a natural choice to empirically validate this. We did not provide more downstream experiments (e.g. semi-supervised performance) as we wanted our message to be simple: if a researcher is looking into hybrid models and finding that coupling blocks aren't working well, they should consider residual blocks (with unbiased estimation).

**Temperature trick for sampling.**  We have looked into this and will update the paper. The temperature trick used by Glow only works when the log-determinant does not depend on the sample itself, i.e. additive coupling blocks. This allows an equivalence between reducing entropy in the Gaussian base distribution and a temperature-annealed model. For general flow models, obtaining quality samples efficiently would be a topic on its own.

**CelebAHQ 64x64.**  We used a downsampled version of CelebAHQ which was also qualitatively used by Flow++.

## Reviewer #2:

**Generalization to other methods/areas.**  We argue that our contributions bring to light interesting technical innovations that can be generalized to vastly different domains and applications, though we agree that the method of application will likely not be straightforward and require much thought.

Many works that make use of the Russian roulette estimator are already referenced in our related work section, which include graphics and optimization. Furthermore, our estimation approach can lead to improvements in existing works that contain infinite series or log-determinants, such as [3] which approximates the density of GAN samples but may have a bias problem. The gradient formulation is derived using the idea of a Neumann series, which was also used in [1] to derive a "recurrent backpropagation" for training recurrent neural nets.

With regards to applicability within the invertible/flow literature, the design of activation functions with meaningful second-order derivatives will be useful for models that actually compute the log-determinant (or any other objective) using differentiation. Currently, research is still focused on models where the log-determinant is computed using the *output* of a neural net rather than the derivative of one.

Thanks to the reviewer's comments, we may include some of this explanation in the introduction itself.

## Reviewer #3:

**Inequality in Appendix.**  Its reason is that the two sides are different infinite series with the same (convergence) value, but different terms. They differ after the unbiased estimators are used. They may have slightly different variances, but we did not notice any significant differences in practice and it's not clear whether one is theoretically always lower variance than the other. We will add more clarifications in the appendix.

**Adaptive SN.**  We always perform power iteration to convergence, so the number of iterations is adaptive. It's usually very high at initialization, then around 1$\sim$5 iterations after every weight update, with the mode being 1.

**References**  (We will add [1-2] to the main text. [3] is already cited.)

[1] "Reviving and Improving Recurrent Back-Propagation" Liao et al. (2018)
[2] "Autoregressive Quantile Networks for Generative Modeling" Ostrovski et al. (2018)
[3] "Backpropagation for Implicit Spectral Densities" Ramesh & LeCun. (2018)


[Meta-Review · NeurIPS 2019]

This paper received very good scores, with strong consensus in the reviews. It makes clear improvements relative to previous residual flow generative models.